# Evaluation of Avian Reovirus S1133 Vaccine Strain in Neonatal Broiler Chickens in Gastrointestinal Integrity and Performance in a Large-Scale Commercial Field Trial

**DOI:** 10.3390/vaccines9080817

**Published:** 2021-07-23

**Authors:** Victor Manuel Petrone-Garcia, Joshua Gonzalez-Soto, Raquel Lopez-Arellano, Mariano Delgadillo-Gonzalez, Victor M. Valdes-Narvaez, Fernando Alba-Hurtado, Xochitl Hernandez-Velasco, Inkar Castellanos-Huerta, Guillermo Tellez-Isaias

**Affiliations:** 1Programa de Doctorado en Ciencias de la Salud y Producción Animal, Facultad de Estudios Superiores Cuautitlán, Universidad Nacional Autónoma de Mexico, Cuautitlán Izcalli 54714, Mexico; castellanos.inkar@gmail.com; 2Departamento de Ciencias Biológicas, Facultad de Estudios Superiores Cuautitlán, Universidad Nacional Autónoma de Mexico, Cuautitlán Izcalli 54714, Mexico; joshuagsotomvz@gmail.com (J.G.-S.); drfernandoalbafesc@gmail.com (F.A.-H.); 3Laboratorio No 5: LEDEFAR, Unidad de Investigacion Multidisciplinaria, Facultad de Estudios Superiores Cuautitlán, Universidad Nacional Autónoma de Mexico, Cuautitlán Izcalli 54714, Mexico; lopezar@unam.mx; 4Independent Poultry Zootechnical, Aguascalientes 20000, Mexico; degomariano@hotmail.com; 5Independent Poultry Nutritionist, Texcoco 56100, Mexico; valdesnv@gmail.com; 6Departamento de Medicina y Zootecnia de Aves, Facultad de Medicina Veterinaria y Zootecnia, UNAM, Coyoacan 04510, Mexico; xhernandezvelasco@gmail.com; 7Department of Poultry Science, University of Arkansas, Fayetteville, NC 72701, USA; gtellez@uark.edu

**Keywords:** reovirus S1133 vaccine, pancreatic and enteric histology, ELISA

## Abstract

Avian reovirus (ARV) is the principal cause of several diseases. The vaccination of breeders allows for the control of viral arthritis and delivery of maternal-derived antibodies to the progeny. The vaccination of broiler chickens with ARV strain S1133 is used to prevent viral arthritis. However, the post-vaccination enteric effects have not been well-characterized. The purpose of this study was to evaluate the effect of vaccination with the S1133 strain on the weight gain and feed conversion of broiler chickens and to characterize the gastric, enteric, and pancreatic lesions that the strain could induce. A total of 672,000 chickens were divided into two groups: a group vaccinated with ARV strain S1133 (S1133ARV) and a control group (not vaccinated). Upon histological analysis, the vaccine group showed less proventricular glandular tissue and atrophy of the pancreas and duodenal villi, as well as having a lower average daily profit. The conclusion based on the results of this investigation is that neonatal vaccination with S1133ARV causes atrophy of the pancreatic acini, proventricular glands, and intestinal villi, leading to an increased diameter of the glandular lumen and atrophy of the enteric villous, as well as weight loss, in broiler chickens.

## 1. Introduction

Avian reoviruses are part of the *Reoviridae* family in the genus *Orthoreovirus*. They are a nonenveloped virus composed of two concentric icosahedral capsids with an external diameter of 80–85 nm [1].

Member viruses have a broad host range, including metazoans, plants, protists, and fungi [2]. As in other species, the virus is abundant in poultry, and most reoviruses are innocuous [1,2,3]. The term “reovirus” is an acronym for “respiratory, enteric, orphan virus” since it was first isolated from the lungs and intestines in humans with no clinical signs [3]. In commercial poultry, pathogenic viruses cause significant economic losses due to arthritis and tenosynovitis in the gastrocnemius tendons [4]. Viral arthritis mainly affects meat-type chickens but has also been diagnosed in commercial layers [5]. Breeder flocks that develop viral arthritis during egg production may be characterized by lameness, increased mortality, decreased egg production, suboptimal hatchability/fertility, and vertical transmission of the virus to progeny [6]. Shedding of virulent reovirus vertically by a breeder flock may affect progeny and cause severe losses. Moreover, since avian reoviruses replicate in the gastrointestinal tract, they are also associated with other pathologies such as stunting malabsorption syndrome, hepatitis, gastroenteritis, myocarditis, and respiratory diseases [3,7,8]. Avian reoviruses possess group- and serotype-specific antigens, and neutralizing antibodies can be detected 7–10 days following infection.

Vaccination against reoviruses in broiler breeders is conducted with live apathogenic vaccines (strain 2177), modified vaccines (strain S1133), and inactivated vaccines produced with pathogenic reoviruses (strains S1133, 2408, SS412, and 1733). In some countries, homologous viruses from the poultry geographic area are also used [1,3,4]. The apathogenic live vaccine and inactivated vaccines are administered subcutaneously, while modified live vaccines are used in drinking water. Vaccination in broiler breeders is essential to protect them against viral arthritis. However, strong vaccination programs in breeders are also used to transfer passive immunity to protect their progeny against viral arthritis [1,9,10]. Neonate chickens are highly susceptible to pathogenic reovirus infection [11,12]. Hence, proper vaccination of broiler breeders is crucial [13,14]. Maternal antibodies can afford protection to 1-day-old chicks against natural and experimental infections, but the level of protection conferred by antibodies is related to serotype similarity, virus virulence, host age, and antibody titer [3]. Recovery from reovirus infection involves both B- and T-cell activity, but protection is predominantly B-cell-mediated (antibodies). Therefore, maternal immunity is essential for protection against viral arthritis [15]. The experimental suppression of T-cell-mediated immunity resulted in increased mortality in reovirus-infected birds, but the relative severity of tendon lesions was unaffected [16]. CD8^+^ T cells may play a major role in pathogenesis and/or reovirus clearance in the small intestine. In this process, maternal immunity does not play an important role [17].

The S1133 avian reovirus strain (S1133ARV) is the most widely used for vaccination, and it has been effective against viral arthritis in most parts of the world [1]. However, as far as we are aware, no evidence of the clinical expectation has been reported on using the live modified S1133ARV strain in neonate broiler chickens under commercial conditions. Hence, the purpose of this study was to evaluate the effect of S1133ARV on the weight gain and feed conversion of broiler chickens following vaccination with this strain, in addition to characterizing the gastric, enteric, and pancreatic lesions induced in response 1-day-old broiler chickens in a large-scale commercial field trial in Mexico.

## 2. Materials and Methods

### 2.1. Application of the Avian Reovirus S1133 Strain Vaccine in Broiler Chickens under Commercial Conditions

#### 2.1.1. Location and Facilities of the Large-Scale Commercial Field Trial

This study was conducted at the regional complex for one slaughterhouse from an integrated poultry producer located in Aguascalientes, Mexico, with a clinical history of previous flocks of diarrhea with undigested feed with orange mucus, twisted pancreas, and a deficit of 4 g·day^−1^ of average daily gain, without arthritis or tenosynovitis. Twenty-four chicken houses with a capacity of 28,000 female broiler chickens were selected (*n* = 672,000 total chickens). Twelve houses were randomly selected, and chickens were vaccinated at 1-day-old using a spray cabinet with the avian reovirus S1133 strain, whereas the other twelve houses served as the nonvaccinated control group. Chickens were raised under normal production conditions and fed a four-phase commercial basal diet [18] (Appendix A). Evaluation of production parameters was done at the end of the grow-out cycle (38 days of age). Chickens were housed in a conventional farm with natural ventilation featuring an age-appropriate environment and kept under ambient conditions using the equipment recommended by the Ross broiler management handbook [19]. Evaluated parameters included the age of the birds at processing, average daily gain (ADG), feed conversion rate (FCR), livability (LI), and production efficiency factor (PEF).

#### 2.1.2. Source of Animals

Female broiler chickens were hatched at the commercial hatcheries of a Mexican poultry company in Aguascalientes, Mexico. The ROSS^®^ 308 chickens (Aviagen^®^, Huntsville, AL, USA) came from imported embryos (Keith Smith Farms^®^, Hot Spring, AR, USA). The reovirus vaccination program for broiler breeders with an active virus was performed, as described, according to age: on day 0 with the 2177 strain (2177^®^, Merck Sharp and Dohme Corp, Kenilworth, NJ, USA); at 2, 4, and 6 weeks with the inactivated virus strain S1133 (Enterovax^®^, Merck Sharp and Dohme Corp, Kenilworth, NJ, USA). Vaccination with the inactivated virus was performed at 12 weeks using the inactive S1133, 2408, and SS412 strains (Maximune^®^ 8, Ceva, Libourne, France), as well as the autogenous strain (custom KV: 9802. Elanco, Greenfield, IN, USA), and at 18 weeks with the S1133 (AviPro^®^ 106 REO or KV: 7805, Elanco, Greenfield, IN, USA), and 1733 strains (AviPro^®^ 106 REO or KV: 7805, Elanco, Greenfield, IN, USA), as well as the autogenous strain (custom KV: 9802, Elanco, Greenfield, IN, USA).

#### 2.1.3. Avian Reovirus S1133 Vaccine

The avian reovirus strain S1133 included in the live modified virus vaccine cloned in tissue culture (Enterovax^®^, Merck Sharp and Dohme Corp, Kenilworth, NJ, USA) was used as the challenge virus with a titer of 10^6.5^ median tissue culture infectious dose (TCID_50_)/mL according to the manufacturer’s recommendation. Chickens in the vaccinated group received a full dosage of avian reovirus S1133 strain [20] at 1-day-old using a spray cabinet (Spra-Vac II^®^, Boehringer Ingelheim Vetmedica^®^, Guadalajara, Mexico).

### 2.2. Performance Variables

The performance variables were calculated as described in this section. The gain weight of the flock (GWF) was measured in kg at slaughterhouse reception divided by the initial number of chicks (INC), excluding mortalities at chick reception. The gain weight of the chickens (GWC) was calculated from the GWF divided by the INC. The ADG (g·day^−1^) was calculated from the GWC divided by the INC. Mortality was calculated based on chickens received at slaughterhouse reception from the IAC. Livability (LI) was calculated by subtracting the mortality from 100. The feed intake of the flock (FIF) was determined as the difference between the total amount of feed offered and the number of refusals. The feed conversion ratio (FCR) was calculated by dividing FIF with GWF. The production efficiency factor (PEF) was calculated using the following equation:$$\mathrm{PEF}=\frac{\mathrm{LI}\;\left(\%\right)\times \;\mathrm{GWC}\;\left(\mathrm{kg}\right)}{\mathrm{Age}\;\left(\mathrm{days}\right)\times \;\mathrm{FCR}\;}\times 100.$$

#### Cost–Benefit Calculation of Vaccination against Avian Reovirus

A cost matrix was built with variable feed intake·chick^−1^ by average feed cost (462.38 USD), according to the feed program and constant other costs (0.82 USD·chick^−1^), estimated using the chicken production cost of the Mexican Poultry Federation (UNA) [21] (Appendix A). The avian reovirus group included an additional cost of 0.0102 USD·chick^−1^. Chick income was obtained in terms of live body weight (kg) by actual price per kg (1.60 USD), as a function of livability. Profiles were calculated as the difference between chicken income and cost [22].

### 2.3. Sample Collection and Processing

Figure 1 shows the methodology flow chart. At 0, 7, 14, 21, 28, and 35 days of age, 10 chickens from each house were blended (*n* = 120) from each group. The blood serum was placed in refrigeration (2 °C). At 14 days of age, one chicken from each chicken house was randomly selected (*n* = 12) from each group, euthanized by cervical dislocation, and necropsied. Samples of each chicken were taken from the middle parts of the proventriculus (PV), pancreas (PA), proximal duodenal branch (PD), and distal duodenal branch (DD), as well as 3 cm caudally to Meckel’s diverticulum in the distal jejunum (DJ) (Figure 2a). The samples were fixed immediately by immersion in 10% neutral buffered formalin. Tissues were then processed and embedded in paraffin using routine histological techniques.

### 2.4. ELISA for Assessment of Reovirus Antibodies

The obtained serum samples were analyzed for antibodies against reovirus using commercially available enzyme-linked immunosorbent assay tests (Reo ELISA CK100^®^, BioCheck^®^ UK LTD, Ascot, UK). The ELISA commercial test is widely used for assessing reovirus antibody levels on a flock basis. The test is efficient for the detection of antibodies to avian reovirus in *Gallus gallus*. The ELISA method was developed using whole virus antigen, as well as recombinant σC and σB; thus, the test disallows the differentiation of infected from vaccinated chicks [23].

### 2.5. Histopathology

Paraffin-embedded tissues were sectioned, mounted, and stained using hematoxylin and eosin (H&E) and examined for lesions; tissues were evaluated by photon microscopy using the AmScope^®^ 3.7 (Irvine, CA, USA) image analysis program. Each tissue was also assigned a lesion severity score. The proventriculus evaluation consisted of measuring the transverse diameter of the total diameter (TG) and the luminal diameter (LG) of the proventricular gland (Figure 2b). Lymphoid nodules contained in the glandular zone of the proventricular mucosa were also counted. The intestinal evaluation considered the PD, DD, and DJ. From the three intestinal zones, the total thickness (TM) and the lamina propria (LP) of the mucosa were measured, in addition to the villous height (VH) (Figure 2c). Cysts and lymphoid clusters contained in the glandular zone of the intestinal mucosa were also counted. Pancreas analysis consisted of obtaining the percentage tissue degeneration and counting the necrotic foci and lymphoid clusters; the fibrosis/acinar atrophy score was also calculated. The lymphoid infiltrate was evaluated using a digital microscope camera with a field of view (FOV) of 3.4 mm^2^ using a 5× objective lens. The largest cross-diameter of the lymphocyte clusters was used to quantify the number of lymphocyte cell layers (Figure 2d). The acinar fibrosis/atrophy score was based on the number of fibroblast layers embedded in bands separating the PA acini (Table 1). The acinar fibrosis/atrophy score was evaluated with a FOV of 0.87 mm^2^ and a 10× objective lens (Figure 2e). The acinar fibrosis/atrophy was obtained from the total of 60 scores calculated (five FOVs for 12 tissue cuts). The number of layers was multiplied by the number of clusters, and the average was obtained from the total of 60 scores calculated (five FOVs for 12 tissue cuts).

### 2.6. Data and Statistical Analysis

The fibrosis/atrophy PA score was analyzed using the Mann–Whitney U test. The remaining data confirmed normal distribution (Shapiro–Wilk test) and homoscedasticity (Levene test). Consequently, the data were subjected to a parametric test (one-tailed Student’s *t*-test). Prior to statistical analysis, the percentage mortality, L/TG, LP/V, and degeneration PA were subjected to an arcsine square root transformation. The statistical significance was set at *p* < 0.05.

## 3. Results

### 3.1. Performance Variables

The results of the performance variables of the female broiler chickens vaccinated with the avian reovirus S1133 strain are summarized in Table 2. Significant reductions (*p* < 0.05) in average daily gain and production efficiency factor, as well as an increase in FCR, were observed in chickens that received the avian reovirus vaccine when compared with nonvaccinated control chickens (Table 2).

#### Cost–Benefit Calculation

The two groups presented the same cost of production (*p* = 0.4789). However, the S1133ARV group presented lower income and profits (*p* = 0.0229 and *p* = 0.0335) than the control group (Table 3).

### 3.2. Histopathology

Table 4 shows histological measurements of the proventricular gland from broilers vaccinated with the S1133 reovirus strain. Both groups had the same TG of the PV (*p* = 0.724), whereas the LG of the proventricular glands of the S1133ARV-vaccinated broilers was higher than that of the control broilers (*p* = 0.017).

The histological measurements of the enteric mucosa from female broilers vaccinated with the reovirus strain S1133 are summarized in Table 5. The VH values from the PD of the control broilers were higher than those of the S1133ARV-vaccinated broilers (*p* = 0.00005). However, both groups had the same VH and TM of the DD and DJ (*p* = 0.075 and *p* = 0.066). The LP of the PD and the DD of the S1133ARV-vaccinated broilers were higher (*p* = 0.00005 and *p* = 0.015) than those of the control broilers. However, both groups had the same LP of the DJ (*p* = 0.365). Both groups had the same TM of the PD (*p* = 0.242), the DD (*p* = 0.189), and the DJ (*p* = 0.123).

Table 6 shows the results of the pancreatic histological evaluation of broiler chickens vaccinated with the avian reovirus S1133 strain. No significant changes were observed in terms of degeneration, necrosis clusters, and lymphoid clusters between both groups (*p* = 0.171, *p* = 0.612, and *p* = 0.060). However, the fibrosis scores of the S1133ARV-vaccinated broilers were higher (*p* = 0.022) than those of the control broilers.

### 3.3. Antibody Titers

The results of the antibody titers from broiler chickens vaccinated with the avian reovirus S1133 strain are shown in Figure 3. Chickens in both groups revealed a high maternal antibody titer against the avian reovirus S1133 strain, which is consistent with the vigorous vaccination program of broiler breeders against reoviruses. In both groups, maternal antibody titers showed a progressive reduction on days 7, 14, and 21 of evaluation. Interestingly, on days 28 and 35, the antibody titers in both groups were increased. However, no differences (*p* > 0.05) in antibody titers were found between the two groups across all weeks of evaluation (Figure 3).

## 4. Discussion

The results of the present study indicate that vaccination of neonatal broiler chickens with the avian reovirus S1133 strain has a negative economic and productive impact since replication of the virus induces pathological alterations in the gastrointestinal tract. The use of the vaccine was not justified in the cost–benefit analysis.

The reduction in performance was associated with histopathologic and morphometric changes in the proventricular gland, duodenum, and pancreas of female broiler chickens that were commercially processed at 38 days of life. These findings agree with previous researchers who reported that avian reovirus isolated from intestinal contents of broiler chickens with malabsorption syndrome produced a transient but significant depression in body weight gain when inoculated orally into 1-day-old chicks [24]. In addition, Jones and Georgiou found that resistance to reovirus is age-related because, although reoviruses can infect older birds, the resulting disease is generally less severe, and the incubation period is longer [6].

The decline in the performance of vaccinated broilers can be explained by histological findings. The luminal diameter of the proventricular gland was significantly larger in the vaccinated group. This increase in lumen was due to a reduction in glandular tissue, which caused the ratio of the lumen over the total glandular diameter radius to increase. As is known, glandular tissues produce hydrochloric acid and pepsinogen, which are essential for the digestion of proteins [25].

In the duodenal villi of the vaccinated group, there was a decrease in villus height and an increase in lamina propria such that the ratio of the lamina propria thickness over the total thickness of the mucosa radius increased. Apoptosis is the process via which reoviruses cause epithelial atrophy of the proventricular glands and the epithelium at the tip of the villi [26,27]. This phenomenon may explain the absence of an evident inflammatory process in the tissues observed in this work [28]. These duodenal changes induce a decrease in nutrients, especially in proteins. The highest absorption of proteins in the duodenum occurs in its proximal part, which is the most strongly affected by the vaccine. While the effect of the vaccine virus on enteric villi was lost in the distal jejunum, the absorption of nutrients was lower [25,29].

The pancreas of vaccinated broilers exhibited moderate fibrosis, but this fibrosis was higher in vaccinated chickens compared with nonvaccinated control chickens. An increase in the amount and density of the interstitial connective tissues with compression atrophy of the acini characterizes the chronic phases of inflammation in the pancreas [30,31,32]. Typically, bands of mature fibrous tissue separate small lobules of acinar tissues [33]. Pancreatic fibrosis is attributed to selenium deficiency in stunted broilers. One of the limitations of the present study is that the concentration of selenium was not measured. However, a reduction in nutrient absorption is associated with selenium deficiency [34]. It is suggested that the normally low activities of selenium-dependent glutathione peroxidase (SeGSHpx) in the pancreas may predispose that organ to atrophy due to oxidative stress under conditions of nutritional selenium deficiency, resulting in further depletion of SeGSHpx [25]. Pancreatic atrophy seems to be dependent on selenium concentration. In the present study, the pancreatic atrophy was moderate, whereas, in the works of Whitacre et al. [35] and Xu et al. [31], the deficiency and atrophy were severe.

The results of the ELISA revealed a high maternal antibody titer, which is consistent with the vigorous vaccination program of broiler breeders against reoviruses. However, the maternal antibodies did not prevent infection of the live modified avian reovirus S1133 strain in the gastroenteric epithelium. Hence, the replication of the S1133 strain caused damage to the proventricular glands and enterocytes. In the present study, maternal antibodies decreased at 3 weeks of age in both groups. Interestingly, an increase in antibody titers was observed in both groups, presumably, due to a wild reovirus challenge that was not controlled by vaccination, as previously reported by Zhong et al. [13]. Reoviruses can be isolated from healthy birds, and serum antibodies are often found in both affected and healthy birds [3]. Another limitation of the present field trial study is the lack of isolation and characterization of the wild strain(s) responsible for inducing an immune response in nonvaccinated control chickens. However, in the present study, performance parameters of nonvaccinated control chickens were not affected, even though they showed antibody titers against avian reovirus. The ELISA commercial kit used to evaluate antibody titers (Reo ELISA CK100^®^, BioCheck^®^) allows for the detection of serum antibodies against all serotypes of avian reovirus in both vaccinated and nonvaccinated flocks; hence, it can be used for screening for field infections, as well as for monitoring vaccination success in poultry. In summary, while the large sample size and per-house randomization schema provide conclusive data regarding the effect of the reovirus S1133 challenge in the present study, limitations to the generalizability of these results compared with other commercial facilities receive short shrift. Further studies to evaluate the use of the reovirus S1133 strain in neonate commercial chickens under different breeder vaccination stratagems that may affect maternal antibody levels or under different background prevalence of ARV infection should be investigated.

## 5. Conclusions

Strong broiler breeder vaccination programs with the avian reovirus S1133 strain are designed to prevent viral arthritis in breeders and the progeny through passive immunity. Since the virus replicates in the gastrointestinal tissue, regardless of the maternal antibodies, the results of the present study suggest that neonatal vaccination in broiler chickens with the live avian reovirus S1133 strain should be avoided, as it leads to a disruption of gastrointestinal integrity and a decrease in performance. Hence, the cost–benefit analysis demonstrated that the use of this vaccine has a negative impact on company profits.

## Figures and Tables

**Figure 1 vaccines-09-00817-f001:**
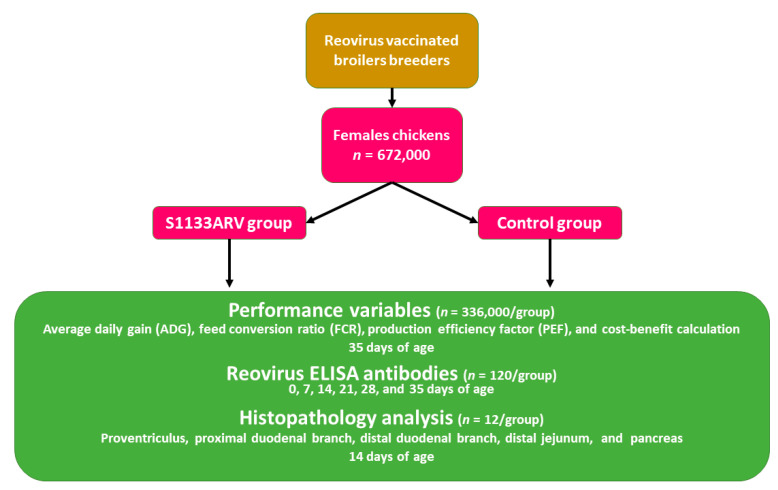
Twenty-four chicken houses with a capacity of 28,000 female broiler chickens were selected from a commercial company (*n* = 672,000 total chickens). Twelve houses were randomly selected, and chickens were vaccinated at 1-day-old using a spray cabinet with the avian reovirus S1133 strain, whereas the other 12 houses served as the nonvaccinated control group. The variables evaluated were productive performance, reovirus ELISA antibodies, and histopathology analysis.

**Figure 2 vaccines-09-00817-f002:**
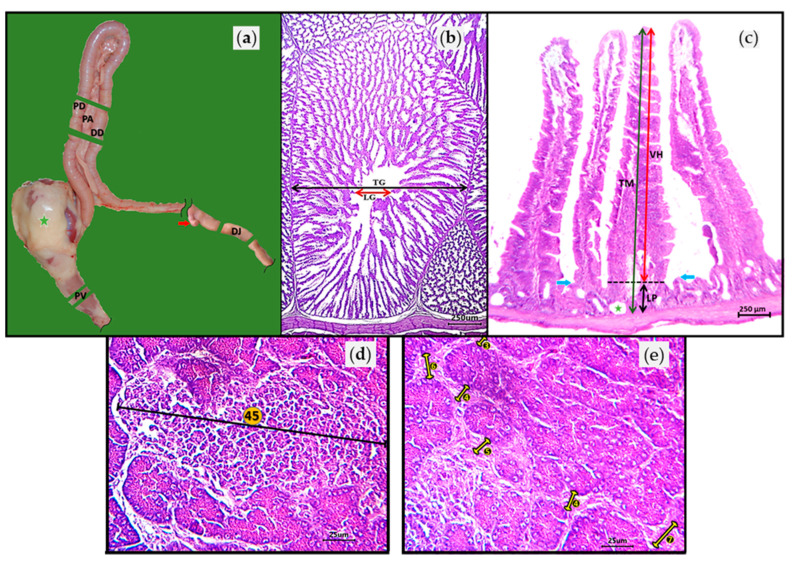
(**a**) Sites where histopathological samples were taken from each chicken from the middle parts of the proventriculus (PV), pancreas (PA), proximal duodenal branch (PD), and distal duodenal branch (DD), as well as 3 cm caudally to Meckel’s diverticulum (red arrow) in the distal jejunum (DJ). Green star = ventricle (gizzard); (**b**) PV consisted of measuring the transverse diameter of the total (TG and black arrow) and luminal (LG and red arrow) diameter of the proventricular gland; (**c**) Intestinal evaluation involved measuring the total thickness (TM, green arrow) and lamina propria (LP, black arrow) of the mucosa, as well as the villous height (VH, red arrow). The segmented line indicates the boundary between LP and VH. The blue arrows indicate the upper pole of the enteric glands that comprise the lintel of the LP. The green star indicates an enteric gland cyst; (**d**) Pancreas histopathologic analysis showing the lymphocyte cluster. The largest cross-diameter (black line) of the lymphocyte clusters was used to quantify the number of lymphocyte cell layers (number in orange circle); (**e**) Pancreas parenchyma. The acinar fibrosis/atrophy score was based on the number of fibroblast layers (number in black circle) embedded in bands separating the pancreas acini.

**Figure 3 vaccines-09-00817-f003:**
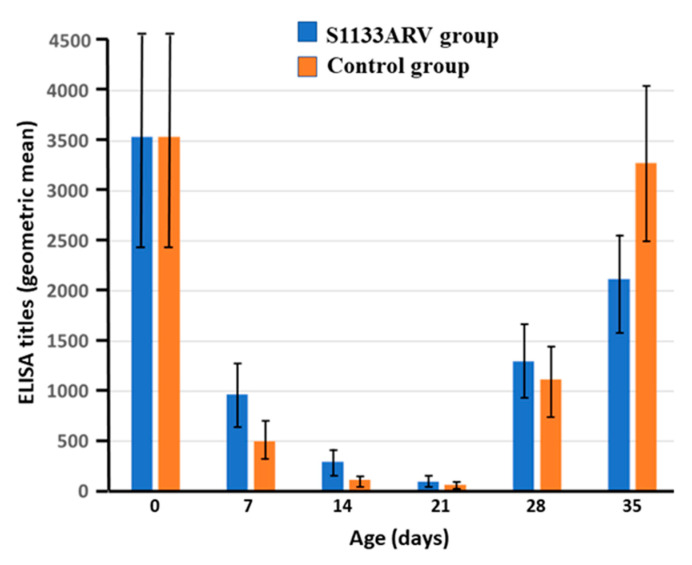
Antibody titers from broilers vaccinated with the avian reovirus S1133 strain (*p* > 0.05).

**Table 1 vaccines-09-00817-t001:** Pancreas acinar fibrosis/atrophy score of the broilers vaccinated with the avian reovirus S1133 strain.

	Percentage (%) of Fibrous Tissue Bands Separating the Pancreatic Acini According to the Number of Fibroblast Layers *		
Score	1 to 2 Layers	3 to 5 Layers	More than 5 Layers
0	0		
0.5	1–5		
1	6–15		
1.5	16–20	1–5	
2	21–35	6–15	
3	36–50	16–20	1–5
4	51–70	21–35	6–15
5	71–85	36–50	16–20
6	86–100	51–70	21–35
7		71–85	36–50
8		86–100	51–70
9			71–85
10			86–100

* The largest number of fibroblast layers observed in the digital microscope camera’s field of view in a single column.

**Table 2 vaccines-09-00817-t002:** Performance variables of female broiler chickens vaccinated with the avian reovirus S1133 strain.

Broiler Groups	ADG (g·day^−^^1^)	FCR	LI (%)	PEF
S1133ARV group	43.46 ± 0.53 ^b^	1.641 ± 0.009 ^a^	95.16 ± 0.35	253.64 ± 3.17 ^b^
Control group	44.82 ± 0.46 ^a^	1.592 ± 0.015 ^b^	94.60 ± 0.54	266.74 ± 4.68 ^a^
*p*-Value	*p* = 0.029	*p* = 0.018	*p* = 0.209	*p* = 0.010

ADG = average daily gain, FCR = feed conversion ratio, LI = livability, PEF = production efficiency factor. Data are expressed as the mean ± standard error. ^a,b^ Different superscript letters within columns indicate a significant difference at *p* < 0.05.

**Table 3 vaccines-09-00817-t003:** Cost–benefit analysis of using the S1133ARV vaccine strain in female broilers.

Broiler Groups	Income (1.60 USD·kg^−1^)	Cost	Profit
S1133ARV group	2.516 ± 0.108 ^b^	2.076 ± 0.051	0.440 ± 0.116 ^b^
Control group	2.581 ± 0.073 ^a^	2.075 ± 0.045	0.506 ± 0.062 ^a^
*p*-Value	*p* = 0.0229	*p* = 0.4789	*p* = 0.0335

The amounts are indicated in USD. Data are expressed as the mean ± standard deviation. ^a,b^ Different superscript letters within columns indicate a significant difference at *p* < 0.05.

**Table 4 vaccines-09-00817-t004:** Histological measurements of the proventricular gland from broiler chickens vaccinated with the S1133 reovirus strain.

Broiler Groups	TG (mm)	LG (mm)
S1133ARV group	1.442 ± 0.285	0.424 ± 0.193 ^a^
Control group	1.144 ± 0.254	0.240 ± 0.114 ^b^
*p*-Value	*p* = 0.724	*p* = 0.017

TG = transverse diameter of the gland, LG = transverse diameter of the lumen of the gland. Data are expressed as the mean ± standard deviation. ^a,b^ Different superscript letters within columns indicate a significant difference at *p* < 0.05.

**Table 5 vaccines-09-00817-t005:** Histological measurements of the enteric mucosa from broiler chickens vaccinated with the avian reovirus S1133 strain.

Duodenal Areas	VH (mm)	LP (mm)	TM (mm)
Proximal duodenum			
S1133ARV group	1.076 ± 0.257 ^b^	0.3235 ± 0.138 ^a^	1.400 ± 0.274
Control group	1.269 ± 0.256 ^a^	0.1845 ± 0.057 ^b^	1.453 ± 0.250
*p*-Value	*p* = 0.005	*p* = 0.00005	*p* = 0.242
Distal duodenum			
S1133ARV group	0.930 ± 0.223	0.2845 ± 0.072 ^a^	1.214 ± 0.247
Control group	1.072 ± 0.357	0.2304 ± 0.086 ^b^	1.302 ± 0.354
*p*-Value	*p* = 0.075	*p* = 0.015	*p* = 0.189
Distal jejunum			
S1133ARV group	0.421 ± 0.203	0.1388 ± 0.075	0.5600 ± 0.247
Control group	0.310 ± 0.147	0.1448 ± 0.043	0.4928 ± 0.144
*p*-Value	*p* = 0.066	*p* = 0.365	*p* = 0.123

VH = villous height; LP = lamina propria thickness, TM = total thickness of the mucosa. Data are expressed as the mean ± standard deviation. ^a,b^ Different superscript letters within columns indicate a significant difference at *p* < 0.05.

**Table 6 vaccines-09-00817-t006:** Pancreatic histological evaluation of broiler chickens vaccinated with the avian reovirus S1133 strain.

Broiler Groups	Degeneration (%)	Necrosis Clusters	Lymphoid Clusters	Fibrosis Score
S1133ARV group	8.18 ± 7.32	0.60 ± 0.89	4.65 ± 5.92	7.82 ± 6.56 ^a^
Control group	9.30 ± 3.20	0.40 ± 0.54	5.95 ± 3.78	2.50 ± 1.98 ^b^
*p*-Value	*p* = 0.171	*p* = 0.612	*p* = 0.060	*p* = 0.022

Data are expressed as the mean ± standard deviation. ^a,b^ Different superscript letters within columns indicate a significant difference at *p* < 0.05.

## Data Availability

Not applicable.

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
