# Peer review of "Evaluation of Avian Reovirus S1133 Vaccine Strain in Neonatal Broiler Chickens in Gastrointestinal Integrity and Performance in a Large-Scale Commercial Field Trial"

_vaccines, 2021, doi:10.3390/vaccines9080817_

Round 1

Reviewer 1 Report

The aim of the authors of this study was to evaluate the administration of the avian reovirus S1133 strain in one-day old broiler chickens in a large-scale commercial field trial in Mexico.

There is an error in the title: replace “gastrointestin12al” with “gastrointestinal”.

The article presents interesting data but improvements need to be made.

The keywords do not identify the topic.

The introduction is well written but could be improved by better establishing the originality of the research aim.

The methods could be improved.

Line 102: replace “f” with “of”.

Figure caption 1 that shows the methodology flow chart is incomplete as it only describes part of the image.

In the paragraph “2.5 Histopathology” only the method used and if the images obtained have been analyzed with image analysis programs should be reported. The figures and the assessments made should be presented in the results.

The caption of figure 2b describes figure 2c and the caption of figure 2c describes figure 2b.

The results should be described in more detail.

Conclusions should be improved without reporting the results again but  placing the research results within a larger context.

Author Response

Reviewer 1:

The aim of the authors of this study was to evaluate the administration of the avian reovirus S1133 strain in one-day old broiler chickens in a large-scale commercial field trial in Mexico.

Dear Reviewer, #1, thank you very much for the time you have spent on reviewing our manuscript. Your comments are very valuable and helpful for revising our paper and guiding our research. We have studied the comments of both reviewers, and the editorial office carefully and have made corrections, which we hope to meet with the approval. Revised portion in the new version were included and are highlighted in yellow in the reviewed manuscript. The following is our point-by-point response to reviewers’ comments:

There is an error in the title: replace “gastrointestin12al” with “gastrointestinal”.

Suggestion accepted.  Thank you. The article presents interesting data but improvements need to be made.

Suggestion accepted, the article has been improving accordingly.  Thank you.

The keywords do not identify the topic.

Suggestion accepted.  Thank you.

The introduction is well written but could be improved by better establishing the originality of the research aim.

Suggestion accepted. Introduction has been modified to better establish the originality of the study. Thank you.

The methods could be improved.

Methods, Figures and Tables have been modified.  Thank you.

Line 102: replace “f” with “of”.

Suggestion accepted.  Thank you.

Figure caption 1 that shows the methodology flow chart is incomplete as it only describes part of the image.

Suggestion accepted.  Detailed description has been added in the legend of Figure 1. Thank you.

In the paragraph “2.5 Histopathology” only the method used and if the images obtained have been analyzed with image analysis programs should be reported.

Tissues were evaluated by photon microscopy using the AmScope® 3.7 image analysis program.  This statement has been incorporated in the text. Thank you.

The figures and the assessments made should be presented in the results.

The Figure 2a to Figure 2E are to illustrate the technique of histological analysis.  Those figures  do not include results.  Hence, we think they should stay in the material and method section and not in the result section.  Thank you.

The caption of figure 2b describes figure 2c and the caption of figure 2c describes figure 2b.

Captions have been corrected.  Thank you.

 The results should be described in more detail.

Suggestion accepted.  Details have been included in the results. Thank you.

Conclusions should be improved without reporting the results again but  placing the research results within a larger context.

Suggestion accepted and the conclusion section has been modified.  Thank you.

Reviewer 2 Report

This manuscript describes a large field trial of avian reovirus S1133 administration to 1-day-old chicks in a real-world setting. The analytical methods are sound and are sufficiently powered. ADG & PEF measurements will be useful to the practitioner. There were missed opportunities to monitor for reovirus infection over the course of the study, to score gait or other factors that may relate to potential positive outcomes of vaccination, and to use chicks with a more appropriate maternal antibody history.

While the large sample size and per-house randomization schema give conclusive data regarding the effect of reovirus S1133 challenge in this study, limitations to the generalizability of these results to other commercial facilities are given short shrift. For example, would the conclusions drawn in this paper hold up under different breeder vaccination strategems that may affect maternal antibody levels? or under different background prevalence of ARV infection? The paper would be improved by a candid discussion of the generalizability to commercial practice. Furthermore, because of the samples analyzed by histopathology, the paper focuses (not wrongly) on gastrointestinal organ systems. Tenosyvitis is scarcely mentioned. Outcomes not captured in the paper’s secondary metrics may be significant contributors to the decision to vaccinate, and the livability index difference could be biologically meaningful. It would be appropriate to mention how factors not captured in this analysis could tilt the cost-benefit calculation.

The article is framed in such a way that the reader unfamiliar with the details of typical avian reovirus vaccination programs may overlook the distinction between the well-established use of S1133 in breeders, i.e. via water vaccination method, and the alternate use in day-old broiler chicks. This is especially true in the conclusion. In addition to editing for clarity, the introduction ought to discuss typical use cases of S1133 and other reovirus vaccines and be explicit about the difference between breeder and broiler vaccination, which is unclear in ll. 70–74 as written.

Clarify the specificity of the Reo ELISA CK100 kit.

Clarify whether the Merck product used is wt S1133 or attenuated by serial passage.

Are most reoviruses really innocuous as stated in l. 47? I would like to see a citation for that.

I recommend putting the sample sizes in the green box as well for Fig. 1 for easy reference of the reader, as the three output measurement types have different n.

Where both primary and derived characters (e.g., LG, TG, and LG/TG) are presented, rather than say which were significantly different and leave it at that (LG and LG/TG), it would be better to discuss whether the difference in ratio came from a change in LG, a change in TG, or both. In this case, LG is clearly the dominant effect while TG holds constant. In Table 4, it appears that a fixed TM exists and varying proportions of that TM are made up of LP versus vili.

Some orthographic errors below. There are several awkwardly-worded or unclear sentences beyond those listed here.

title: gastrointestin12al

  1. 26 AVR → ARV
  2. 42 italicize Reoviridae
  3. 43 family has single-, double-, and triple-shelled particles
  4. 44 family has 9-12 segments
  5. 46 must → most
  6. 47. respiratory, enteric, orphan virus
  7. 60 unclear wording
  8. 76 unclear wording
  9. 91 and throughout: spry → spray
  10. 102 f → of

Fig 2b and 2c are misnumbered in several places

  1. 196 sentence fragment
  2. 232 vaccinated used twice
  3. 248 awkward wording
  4. 252 should have no comma after ‘chickens’
  5. 309 throw → through

Author Response

Reviewer 2:

This manuscript describes a large field trial of avian reovirus S1133 administration to 1-day-old chicks in a real-world setting. The analytical methods are sound and are sufficiently powered. ADG & PEF measurements will be useful to the practitioner. There were missed opportunities to monitor for reovirus infection over the course of the study, to score gait or other factors that may relate to potential positive outcomes of vaccination, and to use chicks with a more appropriate maternal antibody history.

 Dear Reviewer, #2, thank you very much for the time you have spent on reviewing our manuscript. Your comments are very valuable and helpful for revising our paper and guiding our research. We have studied the comments of both reviewers, and the editorial office carefully and have made corrections, which we hope to meet with the approval. Revised portion in the new version were included and are highlighted in yellow in the reviewed manuscript. The following is our point-by-point response to reviewers’ comments:

While the large sample size and per-house randomization schema give conclusive data regarding the effect of reovirus S1133 challenge in this study, limitations to the generalizability of these results to other commercial facilities are given short shrift. For example, would the conclusions drawn in this paper hold up under different breeder vaccination strategems that may affect maternal antibody levels? or under different background prevalence of ARV infection? The paper would be improved by a candid discussion of the generalizability to commercial practice.

Thank you for your comment.  The message we intent to deliver is that strong broiler breeder vaccination programs with the avian reovirus S1133 strain are designed to prevent viral arthritis in breeders and in the progeny through passive immunity.  However, because the virus replicates in the gastrointestinal tissue, regardless of the maternal antibodies, the results of the present study suggest that neonatal vaccination in broiler chickens with live avian reovirus S1133 strain should be avoided as because disrupt gastrointestinal integrity and performance.  This statement has been added in the conclusion section.

Furthermore, because of the samples analyzed by histopathology, the paper focuses (not wrongly) on gastrointestinal organ systems. Tenosyvitis is scarcely mentioned.

This study was carried out at the regional complex for one slaughterhouse from an integrated poultry producer located in Aguascalientes, Mexico, with a clinical history of previous flocks of diarrhea with undigested feed with orange mucus, twisted pancreas, and a deficit of 4 g day−1 of average daily gain, without arthritis or tenosynovitis. This paragraph is in lines 96-100.  Thank you.

Outcomes not captured in the paper’s secondary metrics may be significant contributors to the decision to vaccinate, and the livability index difference could be biologically meaningful. It would be appropriate to mention how factors not captured in this analysis could tilt the cost-benefit calculation.

We appreciate your comment very much.  A cost-benefit analysis has been included in this study and clearly further demonstrate the use of this vaccine has a negative impact in the profit of the company.  

The article is framed in such a way that the reader unfamiliar with the details of typical avian reovirus vaccination programs may overlook the distinction between the well-established use of S1133 in breeders, i.e. via water vaccination method, and the alternate use in day-old broiler chicks. This is especially true in the conclusion.

New paragraphs have been included in the introduction to give more details about the typical avian reovirus vaccination programs in breeders.  The S1133 avian reovirus strain (S1133ARV) is the most widely used for vaccination and has been shown to be effective against viral arthritis in most parts of the world. However, as far as we are aware, no evidence of the clinical expectation has been reported on the use of live-modified S1133ARV strain in neonate broiler chickens under commercial conditions. Hence, the purpose of this study was to evaluate the effect of S1133ARV on the weight gain and feed conversion of broiler chickens following vaccination with this strain in addition to characterizing the gastric, enteric, and pancreatic lesions induced in response in one-day old broiler chickens in a large-scale commercial field trial in Mexico.  Thank you.

In addition to editing for clarity, the introduction ought to discuss typical use cases of S1133 and other reovirus vaccines and be explicit about the difference between breeder and broiler vaccination, which is unclear in ll. 70–74 as written.

Introduction has been modified to clarify the current vaccines programs in the poultry industry.  The use of the S1133 live vaccine is not used in broiler chickens, only breeders.  Thank you.

Clarify the specificity of the Reo ELISA CK100 kit.

Suggestion accepted and the specificity of the Reo ELISA CK100 kit has been clarified.  Thank you.

Clarify whether the Merck product used is wt S1133 or attenuated by serial passage.

The avian reovirus strain S1133 included in the live modified virus vaccine cloned in tissue culture.  This statement has been added to the text.  Thank you.

Are most reoviruses really innocuous as stated in l. 47? I would like to see a citation for that.

Citations have been included.  Thank you.

I recommend putting the sample sizes in the green box as well for Fig. 1 for easy reference of the reader, as the three output measurement types have different n.

Suggestion accepted.  Thank you.

Where both primary and derived characters (e.g., LG, TG, and LG/TG) are presented, rather than say which were significantly different and leave it at that (LG and LG/TG), it would be better to discuss whether the difference in ratio came from a change in LG, a change in TG, or both. In this case, LG is clearly the dominant effect while TG holds constant. In Table 4, it appears that a fixed TM exists and varying proportions of that TM are made up of LP versus vili.

Suggestion accepted and Tables 3 and 4 have been modified.  Thank you.

Some orthographic errors below. There are several awkwardly-worded or unclear sentences beyond those listed here.

title: gastrointestin12al Corrected, thank you.

  1. 26 AVR → ARV: Corrected, thank you.
  2. 42 italicize Reoviridae: Corrected, thank you.
  3. 43 family has single-, double-, and triple-shelled particles: Corrected, thank you.
  4. 44 family has 9-12 segments: Sentence was modified.  Thank you.
  5. 46 must → most: Corrected, thank you.
  6. 47. respiratory, enteric, orphan virus: Corrected, thank you.
  7. 60 unclear wording: Suggestion accepted, and text was modified.  Thank you.
  8. 76 unclear wording: Suggestion accepted, and text was modified.  Thank you.
  9. 91 and throughout: spry → spray: Corrected, thank you.
  10. 102 f → of: Corrected, thank you
  1. Fig 2b and 2c are misnumbered in several places: Corrected, thank you.
  2. 196 sentence fragment: Corrected, thank you.
  3. 232 vaccinated used twice: Corrected, thank you
  4. 248 awkward wording: Corrected, thank you
  5. 252 should have no comma after ‘chickens’: Corrected, thank you
  1. 309 throw → through: Suggestion accepted, and text was modified.  Thank you.

Round 2

Reviewer 2 Report

In the revised draft, the authors have done a good job of describing the specific conditions of this experiment. However, the authors misunderstood the point I was making about the generalizability: I would like to see a couple sentences discussing what factors may impact whether their conclusion will be true in another hatchery.

Table 4 and the economic sensitivity analysis make no sense to include because the cost base enters the model as a constant so of course the control group will perform better at any price per kg from infinity to 0. Because the model used for the sensitivity analysis is inappropriately tautological, it should be removed.

typos include:

vaccinted

Author Response

Dear Reviewer, #2, thank you very much for the time you have spent on reviewing our second draft. We have made corrections to your comments, which we hope to meet with the approval. Revised portion in the new version were included and are highlighted in yellow in the reviewed manuscript. The following is our point-by-point response to reviewers’ comments:

In the revised draft, the authors have done a good job of describing the specific conditions of this experiment. However, the authors misunderstood the point I was making about the generalizability: I would like to see a couple sentences discussing what factors may impact whether their conclusion will be true in another hatchery.

We have added this statement before the conclusion: In summary, while the large sample size and per-house randomization schema give conclusive data regarding the effect of reovirus S1133 challenge in the present study, limitations to the generalizability of these results to other commercial facilities are given short shrift. Further studies to evaluate the use of the revirus S1133 strain in neonate commercial chickens under different breeder vaccination strategems that may affect maternal antibody levels or under different background prevalence of ARV infection should be investigated.

Table 4 and the economic sensitivity analysis make no sense to include because the cost base enters the model as a constant so of course the control group will perform better at any price per kg from infinity to 0. Because the model used for the sensitivity analysis is inappropriately tautological, it should be removed.

Suggestion accepted. Table 4 has been eliminated as well as the economic sensitivity analysis in the manuscript. Thank you. typos include: vaccinated Typo corrected, thank you.
